# The Silent Conquest of *Aedes albopictus* in Navarre: Unraveling the Unstoppable Advance of the Tiger Mosquito Invasion in Progress

**DOI:** 10.3390/insects16080852

**Published:** 2025-08-17

**Authors:** Miguel Ángel González-Moreno, Estrella Miqueleiz-Autor, Itsaso Oroz-Santamaría, Miguel Domench-Guembe, Irati Poveda-Urkixo

**Affiliations:** 1Institute of Smart Cities, Public University of Navarre, 31006 Pamplona, Spain; 2Department of Rural Development and Environment, Government of Navarre, 31005 Pamplona, Spain; irati.poveda.urkixo@navarra.es; 3Institute of Public and Labor Health of Navarre (ISPLN), Department of Health of the Government of Navarre, 31003 Pamplona, Spain; estrella.miqueleiz.autor@navarra.es (E.M.-A.); itsaso.oroz.santamaria@navarra.es (I.O.-S.); miguel.domench.guembe@navarra.es (M.D.-G.)

**Keywords:** adaptation, climate change, vector-borne diseases

## Abstract

*Aedes albopictus*, commonly known as the Asian tiger mosquito, is an invasive species originally from Southeast Asia that has established itself in Europe, including Spain and the Navarrese region. This mosquito is a significant public health concern due to its capacity to transmit viruses such as dengue, Zika, and chikungunya. This study documents the invasion and establishment of *Aedes albopictus* in Navarre, tracing its spread from absence to permanent presence in certain areas. Surveillance is conducted through a network of ovitraps and adult traps, complemented by public awareness campaigns and collaboration with local authorities. Despite ongoing control efforts and information dissemination within the integrated European LIFE-IP NAdapta-CC project, the species continues to expand throughout the region. Eliminating breeding sites remains the most effective method to limit its spread, although complete eradication is unlikely. The tiger mosquito is expected to continue colonizing parts of Navarre in the foreseeable future despite ongoing efforts. Nevertheless, the region is considered to be better prepared to face this challenge.

## 1. Introduction

The impact of climate change on vector-borne diseases is often intricate and multifactorial, occasionally leading to uncertain outcomes [1]. The importance of mosquitoes in the Aedes genus lies in their role as vectors of major arboviruses such as dengue, yellow fever, chikungunya, and Zika [2]. One of these insects is the tiger mosquito, scientifically known as *Aedes albopictus* (Skuse) and commonly named the “Asian tiger mosquito”, which is an invasive exotic species native to Southeast Asia [3].

*Ae. albopictus* is a small black mosquito, measuring between 6 and 9 mm in length. It can be easily identified by a single white line running along the back of its head and thorax. Its legs are also distinctive, being black with white spots. Figure 1 shows an adult specimen (Figure 1).

It is listed both in the Spanish Royal Decree 630/2013 of 2 August, regulating the Spanish Catalogue of Invasive Exotic Species, and in Law 42/2007 of 13 December, on Natural Heritage and Biodiversity [4,5]. Nevertheless, since it is not considered competitive with native fauna or flora and does not displace other species, it is included in public health surveillance and control programs primarily due to its role as a disease vector and the associated health risks.

Due to globalization and human migration, this insect has spread widely across the world despite its limited flight range of 50–200 m. It is considered the most invasive mosquito species worldwide and has spread to Africa, America, Australia and several islands in the Pacific, and Europe [6]. It was first detected in Europe in Albania in 1979 and subsequently spread to countries including Italy (1990), France (1999), Belgium, Switzerland (2003), Hungary, Montenegro, the Netherlands, and Greece (2004) [7,8,9]. In Spain, Catalonia was the first region to report its presence, also in 2004 [8,10]. The detection of the tiger mosquito in 2014 in the Basque Country, a region bordering Navarre, was regarded as an early detection. Intensive surveillance carried out in the area has not prevented the colonization of the Spanish Atlantic slope [11]. In Navarre, its presence was not confirmed until 2018, when an egg was detected and identified, as will be detailed in this paper.

Since 2016, the Institute of Public Health and Labor of Navarre (ISPLN) has been implementing the “Environmental Surveillance Plan for the tiger mosquito (*Aedes albopictus*) in Navarre” [8]. Furthermore, the integrated European LIFE-IP NAdapta-CC project (LIFE16 IPC/ES/000001), focused on climate change adaptation, established a monitoring network that has enabled detailed tracking of the species’ evolution in the region [12,13]. In Spain, other regions have also developed their respective surveillance plans for this species, for example, the Basque Country starting in 2013 [14], Extremadura starting in 2018 [15], and the Region of Murcia in 2019 [16].

Under these circumstances, a new entomological surveillance and control campaign begins every March, which in the last year lasted until December (initially, it was March-October). The mosquito’s expansion is increasingly pronounced due to climate change, which has created favorable conditions not only for its survival but also for its establishment and permanent settlement [16].

This, combined with the tiger mosquito’s role as a vector of emerging and re-emerging viral diseases—especially dengue, but also chikungunya and Zika—makes it a significant threat requiring special attention. Prevention and control are essential to avoid its presence and proliferation and, if necessary, to reduce its population to acceptable tolerance levels.

Despite efforts by public administrations in recent years, the tiger mosquito has expanded and established itself in various municipalities in northern Navarre, mainly in urban and peri-urban areas. Bellini et al. [17] have previously warned about the complexity of controlling and/or eradicating *Aedes* species, which are notable due to their ability to develop in a wide range of artificial breeding sites, mainly on private properties, which significantly hinders entomological control.

The objective of this manuscript is to present the surveillance and monitoring work on the tiger mosquito conducted by public administrations, particularly the ISPLN, in collaboration with the Department of Rural Development and Environment of the Government of Navarre, within the framework of the LIFE-IP NAdapta-CC project and with the support of several local entities in Navarre. This study represents an unprecedented monitoring effort in the region, as it has tracked the progress of this invasive species from the point when it was entirely absent, documented its expansion into multiple areas, and confirmed its definitive establishment in others.

## 2. Methodology

The implementation of the surveillance plan necessarily involves the monitoring of key strategic areas, with the aim of detecting the possible presence of the mosquito before it becomes established. In cases where its presence is confirmed, the objective shifts to designing and implementing an effective integrated control plan. The core prevention and control activities focus on verifying whether the vector is present, managing mosquito populations, and, if necessary, adopting measures to prevent contact with infected individuals.

To this end, an initial agreement was reached during a meeting of the Interdepartmental Commission to place traps in the three areas of Navarre considered at highest risk for the establishment of *Ae. albopictus*, based on population density and traffic patterns. The selected zones were as follows:

The Bidasoa River area, near Irún, Cinco Villas-Bortziriak, extended to the Señorío de Bértiz;The Pamplona region, including the transportation hub, several industrial parks, urban parks, and the vicinity of the municipal cemetery;The Tudela area, including industrial parks, the cemetery, a shopping center, and a public park.

Once the so-called strategic points were defined—based on the risk of introduction through vehicle or human movement, with potential hotspots including tire storage facilities, cemeteries, bus or airport terminals, free trade zones, highway tolls, industrial estates, and shopping centers—the surveillance method was determined in accordance with technical guidelines. For this reason, traps were placed in shaded and vegetated areas, quiet and away from noise, out of reach of people and animals, yet with some human presence, in strategic and discreet locations, near ground level and free from competing attractants.

### 2.1. Types of Traps: Ovitraps and Adult Traps

These traps are described in scientific guidelines [18]. The ovitrap is a simple and cost-effective device designed to detect eggs and occasionally larvae. There are different types of ovitraps worldwide, with a small, dark container capable of storing a certain amount of water being the most common [19,20]. It consists of a container with a capacity of approximately 250–300 mL, filled to three quarters with dechlorinated water, and equipped with a wooden paddle (egg-laying substrate) where mosquitoes deposit their eggs at the air–water interface. Additionally, the rim of the ovitrap is surrounded by a sticky adhesive tape to trap adult mosquitoes as an enhancement (Figure 2A).

The adult trap, specifically the BG-Sentinel-2 model, is more complex. The BG-Sentinel-2 model is one of the trap models widely used in this type of study [21]. It measures approximately 39 cm in diameter and 47 cm in height and weighs around 1.7 kg. Its operation is based on generating a gentle airflow via a fan to disperse an attractant (BG-Lure; Biogents A.G., Regensburg, Germany) placed inside, mimicking human respiration (Figure 2B). The tiger mosquito is attracted by this stimulus and is sucked into a capture net located at the end of a suction tube for later identification in the laboratory. The energy needed to operate the fan was supplied directly from the electrical grid, which was accessible thanks to citizen collaboration. Captured specimens quickly die from dehydration. Figure 2 shows the two trap models used.

The study entitled ‘Vulnerability Analysis and Monitoring Plan’, carried out within the framework of the LIFE-IP NAdapta-CC integrated European project on climate change adaptation for the Navarre region, is the reference document for the placement of the traps distributed throughout the Chartered Community.

This document, only available in Spanish, is public and accessible through the following link: https://lifenadapta.navarra.es/documents/2696321/9392391/An%C3%A1lisis+de+Vulnerabilidad+y+Plan+de+Vigilancia_compressed.pdf (accessed on 3 July 2025).

### 2.2. Environmental Surveillance

Traps were checked on a weekly or biweekly basis depending on the established risk level. For ovitraps, the paddles were collected and placed in labeled bags for delivery to the Agro-food Laboratory of Navarre (LCA, hereinafter referred to as LCA, from the Spanish “Laboratorio Agroalimentario de Navarra”). The water was replaced and a new paddle installed. Adult mosquitoes caught on the sticky tape were collected and transferred into containers. Egg and adult specimens collected from the ovitraps were identified at the LCA. Morphological identification of eggs was conducted under a stereomicroscope. Although *Ae. albopictus* eggs can be morphologically distinguished by their exochorion membrane pattern, they may be confused with eggs of other *Aedes* species [22]. Therefore, molecular analysis via real-time PCR was performed for confirmation. The primers and probes used follow the methodology described by Hill et al. [23]. The specificity of the technique was checked both in silico and experimentally to discriminate from the eggs of *Ae. japonicus*, which is another mosquito species present in Navarre. For adult traps, the capture nets containing the mosquitoes were collected for identification at the LCA, and new nets were installed. Adult mosquitoes were also morphologically identified under the stereomicroscope [19].

### 2.3. Communication and Dissemination

Alongside vector and disease monitoring, ongoing awareness-raising efforts target local authorities, technical personnel, and the general public through various communication channels. It should be noted that each year, the start of the tiger mosquito campaign is announced by sending a press release to the media. In addition, local authorities are kept informed by their political leaders and technical–civil servants through either meetings or direct communications.

## 3. Results and Discussion

The first detection of the tiger mosquito in Navarre occurred in 2018, in the town of Bera. In 2019, a vulnerability analysis of the Navarre territory was conducted, along with the adaptation of the current entomological surveillance plan through external consultancy [24]. Since then, fieldwork has contributed to characterizing the expansion process of this invasive exotic species within just a few years. In some higher-risk areas, control measures such as various entomological blockades using bioinsecticides or registered products (Cipermetrina 12% and terametrina 0.8% for adults and Vectomas FG for larvae) for this purpose were implemented, notably in the Bera area. Bellver-Arnau et al. asserted that preventive treatments yield more optimal results than reactive treatments [25].

Figure 3 illustrates the progression through a sequence of maps of the Chartered Community, showing the affected municipalities, updated and available on the climate change monitoring portal of the LIFE-IP NAdapta-CC project [26] (Figure 3).

The monitoring results obtained so far reveal the species’ establishment in certain areas and its detection in others throughout Navarre. Table 1 summarizes the tiger mosquito monitoring outcomes from 2016 to 2024.

As observed on the map of Navarre, the tiger mosquito entered from the north, at the border with the Basque Country and France, within the municipality of Bera. From there, it has gradually spread to neighboring municipalities to varying degrees. Its initial appearance in the south was recorded in Castejón, a locality also bordering another region, La Rioja. Since 2023, detections in other areas have been sporadic but consistently near geographically significant population centers. The number of municipalities where *Ae. albopictus* has been detected—based on positive egg samples, which do not necessarily indicate the presence of adult mosquitoes—remained at 1 from 2018 to 2021 (except for 2020, when 2 municipalities reported their presence), increasing to 4 in 2022, 11 in 2023, and 18 by 2024. This exponential increase highlights the challenges involved in its containment.

Between 1940 and 2020, according to Swan et al. [27], *Ae. albopictus* spread globally mainly via maritime transport, especially ships carrying used tires. At continental and national levels, passive ground vehicle transport and used tire trade were key drivers, particularly in Europe [27]. From the comparison between the Navarrese areas where the presence of this mosquito has been detected and the main road communication routes of the region, a clear association is evident, particularly regarding the dispersion from the initial sightings to the present. This case invites the consideration that road transport has directly contributed to the movement of specimens from one area to another. This hypothesis gains strength when considering rest areas, driver service stations, logistics centers, or industrial parks located near the sites where detections have occurred.

A particularly notable increase in positive detections of adults and/or larvae was recorded in 2022, a year climatologically ranked as the sixth warmest globally and the second warmest in Europe. In Spain, it was the year with the highest average temperature on record, featuring exceptionally intense, prolonged, and widespread heatwaves during summer [28]. Other studies have already reported and confirm the significance of that exceptionally hot year, which caused the mosquitoes to remain active for longer periods and increased the number of captures during unusual months such as December [29]. The sharp rise in captures during such an unusually warm year suggests, considering climate projections for Navarre, that further expansion and establishment of the tiger mosquito in the region are likely.

Some important aspects are worth highlighting, such as the difficulties in covering the entire geographic area of study and the logistical and operational challenges. The initial approach was a network of strategic monitoring points based on the road network, which has been expanded in some specific areas based on the needs observed at the time. The shortage of technical personnel has also posed a challenge in the implementation of this work, as has the complexity of multiple socio-climatic variables involved: the climate of each area, annual meteorological conditions, the initial lack of technical knowledge, public and citizen awareness of the problem, and so on.

In parallel with the field monitoring, epidemiological surveillance of notifiable diseases transmitted by this vector was conducted. Data were regularly published in the ISPLN Bulletin [30], the results of which are published for free on the ISPLN’s website. To date, all reported cases have been imported (acquired outside Navarre), but continuous monitoring is crucial to prevent contact between infected individuals and the vector.

Regarding communication and outreach, since the start of the LIFE-IP NAdapta-CC project implementation, through which the climate change adaptation strategy has been developed, several meetings have been held with technical staff and political representatives of local entities. Additionally, specific events have been organized to raise awareness of the issue, most notably the June 2024 open-door event. Participation in conferences—via oral presentations, posters, abstracts, etc.—media appearances through interviews and broadcasts, and official press releases from the Government of Navarre, which received coverage in print and digital media, on the radio, and on television, have been other key channels to reach diverse audiences.

Efforts to raise awareness of this health issue among interested stakeholders and/or the general public have not only involved the organization of specific meetings or events, both open to the general public (e.g., LIFE-IP NAdapta-CC’s Open Days) and aimed at technical specialists in the field [31,32,33,34,35,36,37,38,39,40], but also other channels, which are explained below.

As in other Spanish regions, such as Extremadura 2018 [15], the detection of the tiger mosquito has been encouraged through the submission of photographs via the Mosquito Alert, a citizen science application aimed at identifying specimens of the *Aedes* genus in an open and accessible manner [41]. Nevertheless, it can be stated that the impact on Navarrese citizens may not have been as significant as initially desired.

A remarkable milestone has been the creation of a pioneering citizen science initiative aimed at schoolchildren [42], now included in the catalog of citizen science initiatives of the Spanish Citizen Science Observatory [43]. The activity consisted of two parts: one theoretical and the other practical. The theory part was a lecture on the issue of climate change in relation to human health, focusing on the risk posed by vectors. The context of the tiger mosquito and its expansion as an invasive species, its life cycle, and its most relevant characteristics were also explained. Finally, the surveillance program and the traps used were explained, followed by the practical part, which consisted of the placement of ovitraps. Its inaugural edition helped detect the presence of the mosquito in an unmonitored area of Navarre called Sakana [44,45]. Moreover, carrying out this activity has generated additional positive synergies, such as creating impacts in the media and on social networks, which has allowed for greater visibility of the issue at hand. Although the degree of social awareness has not been measurable, an increase is estimated in the areas where the action took place, due solely to transmission from teachers and students to parents, relatives, colleagues, or other individuals in their surroundings.

Furthermore, due to the public health importance attributed to vector risks in the Chartered Community, it is noteworthy that in 2020, a cross-disciplinary technical committee was established under the LIFE-IP NAdapta-CC framework. This was because, in addition to the tiger mosquito, parallel work was being carried out on other vectors, but from the perspective of livestock management [46]. This committee enhances coordination among technical staff, experts, and other stakeholders, creating a unique and possibly pioneering network aimed at a holistic One Health approach. Efforts are currently underway to transform this entity, which lacks any form of official validity or recognition, into a Vector Committee of Navarre formally recognized by the Public Administration [44,45].

Finally, an informational brochure (Figure 4) about the species has been published in both Spanish and Basque [45,47]. Several videos, general to the health sector or specifically focused on the tiger mosquito, have also been produced, some publicly available online with particular mention of the mosquito surveillance and monitoring work. Those aforementioned videos, also available in Spanish and Basque, were uploaded on the YouTube channel of the Department of Rural Development and Environment of the Government of Navarre [48,49]. The preparation of this material was aimed to raise public awareness of the risks posed by this mosquito and to disseminate straightforward measures to prevent its proliferation.

Acknowledging that climate change may lengthen transmission seasons with warmer temperatures and irregular intense rainfall, future extreme weather events could significantly impact tiger mosquito populations and disease risk [50]. As Navarre is increasingly recognized as being impacted by climate change, adaptation efforts within the health sector are now more essential than ever. As a result of all the work carried out in Navarre, the region is considered better prepared to confront *Ae. albopictus* in the future, as well as other invasive exotic species, particularly those of the genus *Aedes*, despite the challenges involved in reducing, halting, and/or preventing their spread.

## 4. Conclusions

The novelty of this manuscript lies in the fact that surveillance was implemented prior to the appearance of the mosquito, with close monitoring carried out in the years following its first detection in the region. It also presents the various actions proposed to address the issue, raise awareness, and attempt to minimize harm from a One Health perspective.

After several years of entomological surveillance, it can be affirmed that the tiger mosquito is now considered established in the northern area of Navarre and its detection is increasing in other areas where it is not yet deemed established. The elimination of potential breeding sites, together with personal protection measures, remains the best preventive strategy to curb its expansion, as eradication appears to be highly challenging. Although some targeted entomological interventions have been implemented sporadically, showing short-term success in controlling rapid spread, the number of captured specimens continues to rise. The species has been observed to disperse beyond the initial detection sites, and its presence is expected to advance throughout much of the region in the coming years.

The significance of this Surveillance Program, in addition to its communication and outreach efforts, underscores the need for continued and expanded collaboration among a growing number of stakeholders. This should positively influence the allocation of resources for decision-making by both political and technical personnel, as well as enhancing public awareness and engagement more broadly.

## Figures and Tables

**Figure 1 insects-16-00852-f001:**
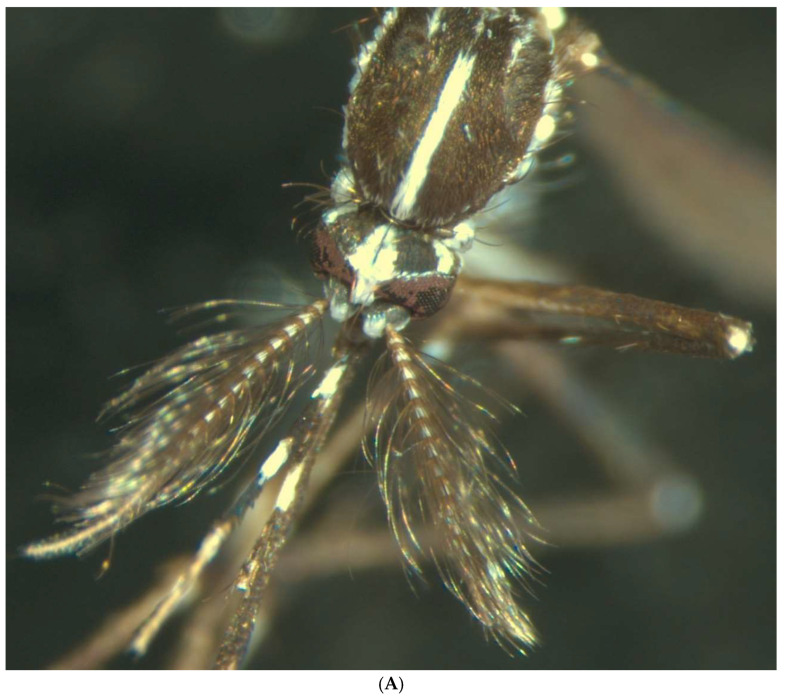
Specimen of the tiger mosquito. The image (**A**) shows a male mosquito, and (**B**) shows a female. Both show the white line on the head characteristic of tiger mosquitoes. In addition, some sexual differentiation can be observed, such as the plumose antennae in the male versus the pilose antennas in the female.

**Figure 2 insects-16-00852-f002:**
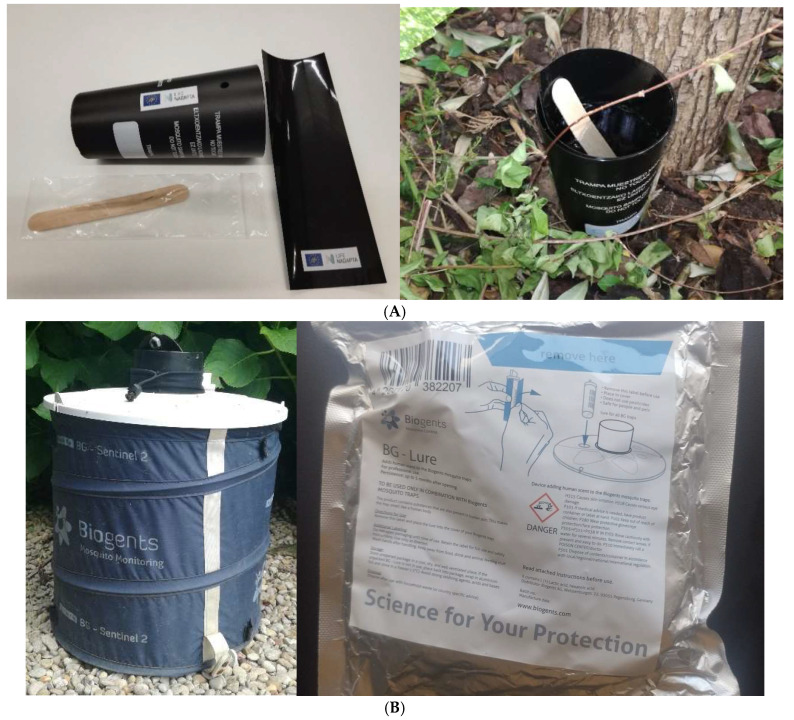
Traps employed. (**A**) One ovitrap dismantled and another already installed. (**B**) An adult trap and the BG-Lure product.

**Figure 3 insects-16-00852-f003:**
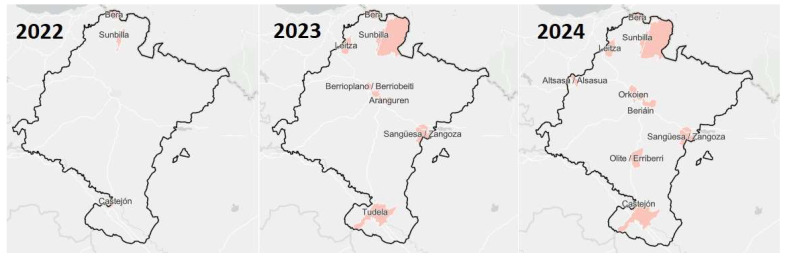
Progression of tiger mosquito expansion across Navarre. Each map shows the affected municipalities where the presence of the tiger mosquito has been detected over the past three years: 2022, 2023, and 2024.

**Figure 4 insects-16-00852-f004:**
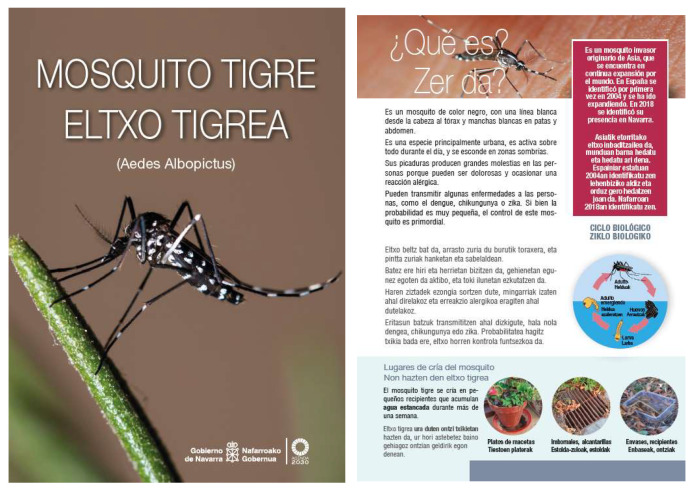
Example of the tiger mosquito informational brochure.

**Table 1 insects-16-00852-t001:** Tiger mosquito monitoring results in Navarre from 2016 to 2024.

Year	Surveillance Points		Samples Collected	Positive Samples		Municipalities Affected	Entomological Blockages Performed
	Adults	Ovitraps		Eggs	Adults + Larvae		
2016	0	47	477	0	0	0	0
2017	0	66	856	0	0	0	0
2018	0	75	817	1	0	1	0
2019	2	53	1359	1	0	1	1
2020	2	62	1654	28	1	2	2
2021	2	62	1617	12	5	1	2
2022	2	63	1745	46	187	4	2
2023	2	67	1817	34	1078	11	2
2024	2	68 + 23 *	1736	29	783	18	1

* The 23 extra ovitraps refer to the traps placed in schools in Navarre that participated in the surveillance and monitoring of the tiger mosquito through the citizen science activity.

## Data Availability

At the monitoring website, data is available: https://monitoring.lifenadapta.eu/ (accessed on 3 July 2025).

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
