# Peer review of "The Silent Conquest of Aedes albopictus in Navarre: Unraveling the Unstoppable Advance of the Tiger Mosquito Invasion in Progress"

_insects, 2025, doi:10.3390/insects16080852_

Round 1
Reviewer 1 Report
Comments and Suggestions for Authors
General comments
González-Moreno et al. present an overview of an Aedes albopictus surveillance program in Navarre Spain. The program was in place before the establishment of Ae. albopictus and document the expansion of the invasive species into the area. The paper provides important documentation of this species’ dramatic range expansion. This manuscript would benefit from more specific explanations of the methods used for surveillance and especially control in the area. Also, from a public outreach perspective were there specific messages used to outreach to the public that can be adapted for use in other areas? More detail should be provided about the school outreach program, including a brief overview of the curriculum presented. This is a current program, but the paper should still be written in past tense.
Specific comments
Make sure species names are italicized throughout, including the references
18- technically the common name is Asian tiger mosquito
36- Only Zika (proper noun) is capitalized, I see it is correct below
49- Not clear what is meant by colonize, aren’t they already in the area? Do you mean continue to expand?
Introduction:
-Ae. albopictus is not just an invasive species in Europe, a brief review of the scope of the problem worldwide would be helpful.
-Add a citation for the climate change statement (line 92-93)
Methods:
-Add a citation for the ovitrap.
-Was BG lure used?
-What is the source of CO2
- It is not clear what the comment about mosquitoes quickly dying from dehydration is leading to.
-How were traps powered?
-How many traps are set?
-Does the sticky tape prevent mosquitoes from laying eggs on the side of the cup?
-What steps are taken to prevent contact? Is increased surveillance performed in response to imported cases? If so, what?
-Please provide more specific information about control interventions. Which biopesticides were used and in what ways
Section 2.4 – this is a very vague description of public outreach efforts.
Results and Discussion:
-Do have any information on levels of insecticide resistance in the invading Ae. albopictus? This may impact the effectiveness of control interventions, especially for residual treatments.
-It would be helpful to include a brief literature review that compares and contrasts the Aedes albopictus invasion and establishment in Navarrese to other areas it has recently expanded into.
-Line 266: impact ‘on’ the Navarrese
-Line 271- not clear what is meant by uncontrolled
-Line 287- remove tab
Figures and Tables:
- Figure 1 – It is an odd choice to use a photo of a male mosquito. Consider adding a picture of a female as well.
-Figure 2 – it would be more helpful to the ovicup in the deployment condition
-Figure 3 – the figure captions are in Spanish and the city names are very small and hard to read.
-Table 1 – the column heading at the far right is just a black box. There are a few typos on the table headings: Ovitraps, and larvae is plural for larva
Author Response
The authors attach a Word document that answers this reviewer's questions, suggestions and other issues.

Reviewer 2 Report
Comments and Suggestions for Authors
To the Authors of Manuscript ID: insects-3789414:
Your manuscript presents a comprehensive study on the invasion of Aedes albopictus in Navarre, Spain, including its detection, expansion, surveillance methods, and control strategies. However, certain aspects of the manuscript could benefit from further clarification or elaboration. Below are some constructive suggestions to improve your work:
- Statistical Analysis:
The quantitative data presented (e.g., egg counts, adult mosquitoes, number of affected municipalities) are presented descriptively. Could you explain the rationale for not performing statistical analyses (e.g., correlations with environmental variables such as temperature or relative humidity)?
- Efficacy of Control Measures:
Although various control measures (e.g., use of bioinsecticides, breeding site elimination) are mentioned, their effectiveness was not quantitatively assessed. Could you clarify whether these omissions were due the descriptive nature of the research or other limitations?
- Communication Strategies:
What methodologies could be applied to assess the impact of communication campaigns (e.g., leaflets, videos, mobile apps) on changes in population behavior?
- Geographical Coverage:
How did the study ensure representative surveillance coverage across Navarre, particularly in remote or hard-to-access areas?
- Study Limitations:
We recommend expanding the discussion on study limitations, including:
- Potential biases in geographical surveillance coverage;
- Logistical challenges in implementing control measures;
- Environmental or operational factors that may have influenced the results.
Author Response

(The authors gave the same response as above.)

Reviewer 3 Report
Comments and Suggestions for Authors
Article
The silent conquest of the Aedes albopictus in Navarre: Unraveling the unstoppable advance of the tiger mosquito invasion in progress
Dear Authors,
Congratulations on your valuable submission. Below, I am sending you my general and specific comments based on the review of your manuscript.
General comments
The manuscript presents a thorough and timely study on the invasion and expansion of the tiger mosquito Aedes albopictus in the Navarre region of Spain. The early implementation of the surveillance plan prior to the official detection of the vector is particularly commendable, allowing for detailed and continuous monitoring of its spread, as well as integration with public policies and awareness campaigns. The comprehensive approach combining entomological surveillance, vector control, and social communication within the framework of the LIFE-IP NAdapta-CC project provides a multidimensional perspective that is crucial for managing invasive species of public health relevance.
The results clearly demonstrate the geographic progression of the mosquito, factors associated with its dispersion, and the impact of climatic variables, strengthening the understanding of its dynamics in the context of climate change. Additionally, the manuscript offers valuable insights into interdisciplinary collaboration and citizen participation strategies, enhancing visibility and social awareness.
To further improve the manuscript, it is suggested to clarify some methodological aspects, especially regarding the definition and justification of the so-called "higher-risk" areas, as well as to enhance the graphical presentation of data to facilitate visual interpretation by readers. A deeper critical analysis of the limitations of the surveillance and control program and a discussion of possible future directions for research and integrated management would also be beneficial.
Overall, this work constitutes a valuable and timely contribution to the knowledge of Aedes albopictus ecology and control in Europe, with direct relevance to public health and climate change adaptation. Its publication would represent a significant contribution to the scientific community and public health policymakers.
Point-by-point comments
Title
Line 2. Please rewrite Aedes albopictus in italic throughout the manuscript. This is a standard taxonomic convention that should be applied consistently in all sections, figures, tables, and references.
Abstract
Background:
Line 34. Rewrite the sentence “currently established in Europe, Spain, and Navarre” to avoid implying that Spain is not part of Europe. A clearer alternative could be: “currently established in Europe, including Spain and the region of Navarre.”
Lines 35-36. Rewrite the sentence “its ability to transmit viruses causing diseases such as Dengue (DENV), Zika (ZIKV), and Chikungunya (CHIKV)” for accuracy. A clearer alternative could be: “its ability to transmit the viruses of dengue (DENV), Zika (ZIKV), and chikungunya (CHIKV) viruses, which cause diseases in humans”. Note that only Zika is a proper name and should be capitalized.
Line 36. Avoid abreviations if they are not used later (DENV, ZIKV,CHIKV).
Methods:
Lines 40-43. Consider rewriting both sentences in the conventional past simple tense. Please clarify whether this refers to ongoing systematic monitoring; if so, make it explicit.
Additionally, the mention of the LIFE-IP NAdapta-CC project should be moved to the Methods section, as it describes the framework and procedures rather than results.
Results:
Line 44. Move “Monitoring within the LIFE-IP NAdapta-CC Project” to methods.
Line 44. Change “the species” by “Aedes albopictus” for greater clarity in this manuscript instance.
Keywords:
Line 51. Avoid repeating title words “Aedes albopictus” and “Navarre”.
Introduction
Lines 54-55. It is unclear why the introduction begins with climate change, a concept that is neither clearly defined nor revisited later in the manuscript. If you choose to keep this focus, please elaborate on the role of climate change in the study context. Additionally, consider using more precise terminology—such as "multifactorial" instead of "multifaceted"—and clarify what is meant by "uncertain outcomes."
Line 56: Consider deleting “and Aedes-borne diseases” from the sentence to improve clarity and avoid redundancy.
Line 62: Please cite Figure 1 in parentheses at the end of the preceding sentence, and apply this citation style consistently throughout the manuscript.
Line 64. Figure 1. Consider a better description of the proposed figure.
Line 74. Add a cite.
Line 92: Please clarify the time frame in the sentence “a new entomological surveillance and control campaign begins every March.” Specify the end month or period to clearly define the surveillance season.
Line 92-94. Add a cite.
Line 96. See comments for line 36.
Line 97. Avoid repetitive use of the term “prevent” in the same sentence.
Lines 133-137. Could you be more specific by providing a clearer description of the criteria used to select the strategic points and/or citing the technical guidelines that were followed?
Lines 197-200. Please specify which registered products and bioinsecticides were used, and provide a justification for the use of these preventive chemical methods, considering that the recommended approach for Aedes albopictus control prioritizes the elimination of containers as potential breeding sites, especially when there is no confirmed virus circulation. Additionally, please define what is considered a "risk area" in this context.
Line 205. Figure 3. Please include a scale bar and a north arrow, clarify the temporal scale represented in each of the three maps, and specify which variable is highlighted in color — for example, whether it indicates the presence of the vector. It would also be useful to display all monitoring points on the map, so readers can interpret the sampling effort and identify sites that consistently tested negative for vector presence throughout the monitoring period.
Line 213. Table 1. The label of the last column is missing or not visible; please ensure it is clearly shown to facilitate interpretation of the data.
Line 219. We suggest placing the comment about Figure 3 immediately after the figure itself and before the in-text reference to Table 1 to improve clarity and maintain a logical flow in the manuscript.
Author Response

(The authors gave the same response as above.)

Reviewer 4 Report
Comments and Suggestions for Authors
This study reports the first findings of the Aedes albopictus in the Navarre region, Spain.
The authors carried out field sampling and identified the collected specimens both morphologically and using molecular techniques. An awareness campaign supported the authorities in implementing preventive measures.
In general, the data reported on Aedes albopictus is of interest and adds new knowledge to the distribution and spread of this invasive mosquito species.
However, the data are reported inadequately. Some important information is missing, while other unnecessary aspects are described in detail.
I suggest the authors read similar studies, such as Goiri et al., 2020 (your reference n.13) and report the data in a similar manner, including tables and figures.
In my opinion, this manuscript is not acceptable for publication.
However, I encourage the authors to completely rewrite the paper, paying attention to form and scientific accuracy, following the example of similar works, because the data deserve to be published.
Below are some suggestions to help with the rewriting process.
In M&M, describing what a tiger mosquito looks like (with a photo) or ovitraps or BG-sentinel is pointless. Instead, other important information is missing, such as: where were the traps placed (with their coordinates)?, what spatial criteria were used (i.e., how far apart were they from each other)?, when did they work (month of the year)?, etc.
Other information is unclear or incorrect, such as:
line 136-137. “the surveillance method was determined in accordance with technical guidelines”. Which ones?
line 150-152. “Its operation is based on generating a gentle airflow via a fan to disperse an attractant (carbon dioxide) placed inside, mimicking human respiration”. Are you sure that CO2 was used as an attractant and not an odorous lure (no source of CO2 is shown in the figure)? If so, specify how the CO2 was supplied.
line 175-177. “Morphological identification of eggs is conducted under a stereomicroscope. Although Ae. albopictus eggs can be morphologically distinguished by their exochorion membrane pattern, they may be confused with eggs of other Aedes species [18]”. The morphological identification of eggs is only possible with special devices, as described in reference 18, and not with any stereomicroscope; therefore, the eggs cannot be distinguished morphologically. For species identification is necessary to hatch the eggs and identify the larvae or adults, or perform a molecular analysis. In this regard:
line 177-179. “Therefore, molecular analysis via real-time PCR is performed for confirmation. The primers and probes used follow the methodology described by Hill et al. [19]”. More details are needed; however, this method is applied to distinguish among Ae. albopictus, Ae. scutellaris and Ae. aegypti. What if there were eggs from other invasive species, such as Ae. japonicus or Ae. koreicus?
Line 181-182. Which morphological keys were used?
And where do the larvae come from? How were they sampled?
Section 2.3 is out of context and does not add any new information to the study. To be removed.
Some statements in the introduction should be moved to the discussion, i.e. from line 90 to 105.
The maps (Figure 3) and Table 1 should be revised.
The maps must show the reference year, the sites where the traps were placed distinguishing between negative and positive for Ae. albopictus presence (and legends not in Spanish).
Table 1 should report:
ovitraps: years, sampling sites, samples collected, positive ovitraps (%), positive municipalities (and “not affected”)
adult traps: years, sampling sites, samples collected, positive ovitraps (%), positive municipalities.
Why are control measures and preventive treatments mentioned if the topic is tracking the presence and distribution of Ae. albopictus?
As far as communication and dissemination are concerned (page 12), the narrative is somewhat overstated.
References must be accessible (e.g., articles presented at conferences must be included at least in the book of abstracts) and, if possible, in English. Too many references to local publications/reports/documents (including many self-citations). Some references are inappropriate (i.e., 2,4,5,7,11,12,37,40,41) others are missing (as in line 70,74,137,182).
Other errors are scattered throughout the text, for example:
scientific names must be in italics
repeated statements (i.e line 118-119 and 119-121)
Comments on the Quality of English LanguageThe English language needs to be improved (check verb tenses).
Author Response

(The authors gave the same response as above.)

Round 2
Reviewer 1 Report
Comments and Suggestions for Authors
The authors have addressed all my comments adequately.
Author Response
According to this reviewer, we have addressed all my comments adequately.
Reviewer 4 Report
Comments and Suggestions for Authors
The authors have only taken a few suggestions into consideration without making substantial changes as requested.
Since the authors have not made the requested changes, in my opinion, the manuscript cannot be accepted.
Author Response
The authors have finally made the changes requested by this reviewer, in accordance with the editor's instructions, pending validation for publication of the manuscript.